# Neuropathology of the Basal Ganglia in SNCA Transgenic Rat Model of Parkinson’s Disease: Involvement of Parvalbuminergic Interneurons and Glial-Derived Neurotropic Factor

**DOI:** 10.3390/ijms231710126

**Published:** 2022-09-04

**Authors:** Emanuela Paldino, Vincenza D’angelo, Mariangela Massaro Cenere, Ezia Guatteo, Simone Barattucci, Giorgia Migliorato, Nicola Berretta, Olaf Riess, Giuseppe Sancesario, Nicola Biagio Mercuri, Francesca Romana Fusco

**Affiliations:** 1Laboratory of Neuroanatomy, Fondazione Santa Lucia IRCCS, 00143 Rome, Italy; 2Department of Systems Medicine, University of Rome Tor Vergata, 00133 Rome, Italy; 3Department of Experimental Neuroscience, Fondazione Santa Lucia IRCCS, 00143 Rome, Italy; 4Department of Motor Science and Wellness, Parthenope University, 80143 Naples, Italy; 5Institute of Medical Genetics and Applied Genomics, University of Tübingen, 72076 Tübingen, Germany

**Keywords:** striatum, rat, Parkinson’s disease, substantia nigra, parvalbumin, trophic factors

## Abstract

Parkinson’s disease (PD) is a neurodegenerative disease characterized by the accumulation of alpha-synuclein, encoded by the *SNCA* gene. The main neuropathological hallmark of PD is the degeneration of dopaminergic neurons leading to striatal dopamine depletion. Trophic support by a neurotrophin called glial-derived neurotrophic factor (GDNF) is also lacking in PD. We performed immunohistochemical studies to investigate neuropathological changes in the basal ganglia of a rat transgenic model of PD overexpressing alfa-synuclein. We observed that neuronal loss also occurs in the dorsolateral part of the striatum in the advanced stages of the disease. Moreover, along with the degeneration of the medium spiny projection neurons, we found a dramatic loss of parvalbumin interneurons. A marked decrease in GDNF, which is produced by parvalbumin interneurons, was observed in the striatum and in the substantia nigra of these animals. This confirmed the involvement of the striatum in the pathophysiology of PD and the importance of GDNF in maintaining the health of the substantia nigra.

## 1. Introduction

Parkinson’s disease (PD) is a neurodegenerative disease characterized by the accumulation of alpha-synuclein, encoded by the *SNCA* gene [1]. The well-known PD triad of symptoms is characterized by tremor, rigidity, and bradykinesia, and is linked to neuronal loss in the substantia nigra pars compacta. Indeed, the pathognomonic feature of PD is the degeneration of these dopaminergic neurons leading to striatal dopamine depletion. Another essential feature of PD neuropathology is the Lewy pathology, consisting of the presence of Lewy bodies and Lewy neurites [2,3] in neurons.

PD is also characterized by cognitive disturbances such as executive functions impairment, language deficits, visuospatial/visuoconstructive disturbance, and impulse control disorders [4,5,6,7,8]. These non-motor symptoms have been associated with a disruption of frontal-subcortical networks [9]. Moreover, mood disorders often occur in PD patients and are thought to derive from extra-nigral pathology. The latter could constitute the reason for the resistance of some cases to pharmacological dopaminergic treatment [10,11,12,13,14].

In 1993, Lin and coworkers described a specific DA neurotrophic factor that is produced by glial cells in rats, namely, the glial cell line-derived neurotrophic factor (GDNF) [15]. This discovery opened new possibilities for the pathogenesis and therapy of PD.

The survival of adult nigrostriatal dopaminergic neurons is strictly dependent on striatal production of GDNF. GDNF is a trophic factor that has been proven beneficial for nigrostriatal neurons, as it displayed significant neurotrophic effects on mesencephalic dopaminergic neurons as well as on motor neurons [16,17,18,19].

It has recently been demonstrated that the population of parvalbumin containing GABAergic interneurons is responsible for the production of most of the striatal GDNF [19]. Indeed, the authors showed that approximately 95% of GDNF-expressing cells in the striatum are PV-positive interneurons, whereas the vast majority of striatal neurons that receive a strong dopaminergic innervation, namely the projection medium spiny neurons, are devoid of GDNF.

PV interneurons are represented throughout the striatum, and they are distributed in a ventral-to-dorsal, medial-to-lateral, and caudal-to-rostral gradient of increasing density. Their distribution coincides with the area innervated by dopaminergic terminals from the substantia nigra [20]. PV interneurons receive dopaminergic inputs from the substantia nigra and glutamatergic synapses from thalamus and cortex. Their axons generate proximal inhibitory synapses onto medium spiny neurons. As in other areas of the central nervous system, (i.e., cortex or hippocampus), striatal PV+ neurons are “fast-spiking” cells that can sustain high-frequency action potential firing with little afterhyperpolarization or spike frequency adaptation [21]. PV cells create a unique network that is interconnected by electrical synapses thanks to dendrodendritic gap junctions, enabling them to fire almost synchronously.

In an in situ hybridization study, cholinergic interneurons of the striatum were found to express GDNF mRNA [22]. Cholinergic interneurons play a key role in modulating the activity of striatum and in regulating dopaminergic and cholinergic signaling [23]. Interestingly, cholinergic interneurons are relatively resistant to several excitotoxic insults [24].

Braak and coworkers described the neuropathology of PD and its progression and processed it into a staging scheme [6]. The striatum could be affected, but in very late stages [25].

Several transgenic models of PD are currently in use. After disease-causing genes were identified using linkage analysis and association analysis with familial and sporadic cases of PD, *SNCA* (α-Synuclein) [26,27] and several other familial PD-linked genes were described, such as parkin, DJ-1, PINK1, and LRRK2 [28,29,30].

In this study, we investigated neuropathological changes occurring in the striatum of a transgenic rat model of PD that overexpresses the human physiological α-syn. This model develops behavioral and clinical changes that recapitulate the disease, such as novelty-seeking, avoidance, and smell, as well as, later on, motor deficits. The neuropathological changes observed in this model involve the integrity of dopaminergic system [31]. In this view, we aimed at studying the striatum and its relevance to the disease.

The evaluation of GDNF distribution in the striatum, as related to neuronal degeneration in the rat PD model, was our interest, given its role in the survival of adult nigrostriatal dopaminergic neurons. We studied striatum morphology and the specific subsets of neurons possibly involved in the GDNF role in PD. Thus, if our hypothesis was correct, a depletion in PV GABAergic interneurons would result in a downregulation of GDNF, which in turn would facilitate the degeneration of substantia nigra.

## 2. Results

### 2.1. Striatal Neuronal Cell Counts

Nissl staining performed on serial sections of rat brain tissue showed a significant neuronal depletion in the 12-month-old *Snca^+/+^* rats (Figure 1A–D). The analysis carried out on the basis of immunofluorescence experiments using a specific marker for the medium spiny projection neurons, Calbindin, showed medium spiny neurons classically distributed in a patch and matrix compartment [32]. However, a significant numerical reduction in Calbindin-positive neurons was observed in adult rats, mainly in the dorso-lateral and ventral part of striatum (Figure 1L). In the wild type rats, the average number of Calbindin-positive neurons was 80+/−10, and this neuronal population was also observed in the 5-month-old *Snca^+/+^* (Figure 1E,F). Over time, a statistically significant reduction in medium spiny neurons manner was observed in a genotype-dependent in the 12-month-old *Snca^+/+^* rats. The statistical analysis, two-way ANOVA performed for the immunofluorescence intensity of the marker Calbindin, detected a statistically significant reduction in animals at 12 months of age (Figure 1P).

### 2.2. Interneuron Subtype Distribution

Medium spiny projecting neurons (MSNs) represent 90–95% of the rodent striatal neurons, and the remaining cells are a mixed population of interneurons [33,34]. Among these, a small percentage (0.3% of all striatal neurons) comprises cholinergic interneurons, and the rest are GABAergic interneurons. GABAergic interneurons are further divided into several neurochemically distinct classes. Approximately 0.7% of all striatal neurons are interneurons that express the calcium binding protein PV [28]. Interestingly, the number of ChAT neurons remained almost unchanged in the WT rats at 5 and 12 months of age, as well as in *Snca^+/+^* rats at the same time points (Figure 2E,F). Conversely, we observed a statistically significant loss the PV+ GABAergic interneurons over time only in the *Snca^+/+^* rats (Figure 3D). Five-month-old WT and *Snca^+/+^* animals showed many PV+ GABAergic interneurons in the dorsal-lateral part of the striatum, where their ontogenic maturation takes place (Figure 3A–C). This condition persisted in the 12-month-old WT rats, while it drastically changed in the 12-month-old rats, whereby most of the PV+ GABAergic interneurons died, mainly in the dorso-lateral part of the striatum. The few surviving PV+ interneurons showed a significantly reduced intensity of immunofluorescence, and a smaller soma with fewer arborizations (Figure 3E,F) (Appendix A).

### 2.3. GDNF Expression in the Snca^+/+^ Rats’ Striatum

For the first time, the distribution of GDNF expression in the striatal neurons was evaluated in the SNCA genetic model of PD, with a focus on medium spiny projection and ChAT and PV+ GABAergic interneurons. Data obtained by analysis of immunofluorescence images revealed the gradual increase in GDNF between 5- and 12-month-old rats, pointing out the importance of this neurotrophic factor for the neuronal maturation. The immunohistochemistry reaction highlights a specific expression of GDNF in the PV+ GABAergic neurons (Figure 4A,B). Specifically, the immunofluorescence intensity analysis revealed the expression of GDNF in MSNs (Figure 4E–G) and ChAT interneurons, where the accumulation of GDNF takes on the appearance of puncta deposits (Figure 4I–K). Subsequently, we evaluated what happens to the PV+ GABAergic interneurons, whose ability to produce GDNF is known. For the first time, we observed a significant colocalization of GDNF in the PV+ GABAergic interneurons in the 5- and 12-month-old rats, which is only retained in the 12 adult WT (Figure 5G), while it is reduced in the adult *Snca^+/+^*, suggesting that the absence of this factor contributes to the specific death of these neurons (Figure 5L). This correlates with the presence of resistant PK aggregates and the increase in the phosphorylated form of human synuclein expression, the function of which could be to alter the ability of PV+ GABAergic interneurons to produce GDNF (Appendix A).

### 2.4. Altered GDNF Protein Expression in the Substantia Nigra

In young rats, we observed an intense immunohistochemical staining of GDNF in the SN pars compacta and reticulata. However, the intensity of staining statistically decreases only in the 12-month-old *Snca^+/+^* rats, in which a reduction in positive cells for GDNF was observed (Figure 6).

## 3. Discussion

The striatum is a major component of the basal ganglia, and is involved in controlling motor activity and reward behavior [35,36]. Different cell types compose the striatum, where 90–95% are GABAergic medium spiny neurons (MSNs), and the remaining 5–10% is constituted by interneurons [37,38]. Striatal interneurons exert a powerful pre- and post-synaptic modulation of striatal functions [39], which implicates them in many movement and psychiatric disorders [40,41].

In this study, we investigated changes in the striatum of a transgenic rat model pf PD. We observed a marked neurodegeneration of the dorsolateral part of the striatum, with a relative preservation of the striosome (patch-matrix) distribution [42] in the surviving areas.

DA projections from the substantia nigra innervate both patch and matrix [43,44], even though matrix is enriched with DA and patch is not. Additionally, the two subclasses of medium spiny neurons, the D1 Receptor and D2 Receptor expressing projection neurons, can reside in either the striosome or matrix compartment; therefore, both compartments contribute to the direct and indirect output pathways. PD is characterized by the progressive degeneration of nigrostriatal pathway, consisting of dopaminergic neuronal bodies of the substantia nigra and their projections to the striatum. Thus, the observation of unchanged patch–matrix compartments in this model could be explained by the notion that the degeneration primarily derives from substantia nigra, which evenly innervates both patch and matrix.

As mentioned above, we observed a marked neuronal degeneration in the dorsolateral part of the anterior striatum. The reason for this dramatic loss of neurons in the striatum in a model of PD is yet to be elucidated. However, striatal neuronal degeneration was previously observed in the MPTP/3-NP model of multiple system atrophy [45], and in PD patients [46,47]. Thus, we can speculate that striatal pathology occurs in the later stages of the disease, at least in this model of PD. From a broader perspective, if striatal degeneration precedes the death of SN, in could be inferred that an altered function of DA neurons would provoke the degeneration of their striatal target.

Several cerebral pathways and circuits that could be involved in PD degeneration have emerged over the years. An imbalance in the activity of cerebellar pathways has been reported in PD [48], such as cerebello–thalamo–cortical circuitry [49]. In a voxel-based functional connectivity analysis, Shen and co-workers (2020) showed an alteration in the connectivity of basal ganglia-cortical circuit in patients with PD that was related to postural instability and gait difficulty. They also suggested, in PD patients with predominant tremor and lesser postural instability and gait impairment, a relation with increased functional connectivity between putamen and cerebellum [50].

Neuronal degeneration involves multiple regions of the brain beyond substantia nigra and VTA, such as locus caeruleus, dorsal raphe nucleus [51], and the dorsal motor nucleus of the vagus. Thus, common features explaining neuronal vulnerability to PD have been investigated. Axonal arborization size, iron content, and autonomous pacemaking activity are some of the biological characteristics that have been implied with cell vulnerability [52,53,54]

In our study, neuronal loss involved mainly the calbindin positive medium spiny projection neurons, along with a subset of interneurons, namely, the parvabuminergic interneurons.

In the striatum, PV interneurons primarily downregulate the activity of medium spiny projection neurons by means of a monosynaptic inhibition, with a feed-forward mechanism [55,56]. In SN, excitoxicity and oxidative stress lead to raised intracellular calcium levels. In this situation, an increase in PV expression was protective [57].

Here, we observed a time-sensitive degeneration of striatal parvalbuminergic neurons that was present in 5-month-old *SNCA* transgenic rats and dramatically worse at 12 months. Other interneurons were relatively preserved, while medium spiny projection neurons decreased progressively. Thus, it appeared that PV neurons were the most vulnerable to the degeneration arising from nigral disfunction due to the *SNCA* mutation.

PV neurons are evenly distributed throughout the striatum, where they provide a homogenous trophic support to the richly ramified axons of nigrostriatal DA neurons. Part of this important trophic function is exerted through the production of GDNF that is retrogradely transported from the DA nerve terminals to the somata located in the SN [58].

In the striatum, GDNF protein expression was decreased markedly in both projection neurons and in degenerating PV interneurons. GDNF protein appeared to be in the neuropil of SNCA^+/+^ animals, possibly deriving from other sources [59]. Indeed, in several disease models, neuroinflammation can upregulate GDNF expression in activated astrocytes [60,61].

PV interneurons are indeed the main source of striatal GDNF [62,63]. Cholinergic striatal interneurons are also a source of GDNF, and we observed that this population was relatively spared by the degeneration of neurons in this disease model.

GDNF displays beneficial effects with respect to the survival of mesencephalic dopaminergic neurons and on noradrenergic neurons of the locus coeruleus [64]. Neuroprotective effects of intrastriatally administered GDNF were observed in animal models of PD [65,66,67]. Moreover, the effect of intrastriatal GDNF administration has been tested in several studies performed in PD patients. However, the clinical benefits of GDNF-based therapies have not been demonstrated [68]. Alternative, indirect GDNF-based therapies have been tested over the years, such as lentivirus (LV) vector transgenes fused with a destabilizing domain [69].

We observed a dramatic decrease in GDNF immunostaining in the dopaminergic neurons of substantia nigra (Figure 6). Thus, our hypothesis that PV interneurons are involved in GDNF signaling in this model of disease was confirmed. Transport of GDNF occurs through a retrograde transport [70].

Our data suggest that the damage at the level of PV+ GABAergic interneurons in the striatum not only determines a reduction in the production of GDNF, but also severely compromises their intrinsic ability to retrogradely transport GDNF to the SN, thus promoting the neurodegenerative process.

Therefore, decreased GDNF availability translates into failure by DA nerve endings to take GDNF from the striatum and transport it retrogradely to the SN, thus contributing to the degeneration of SN. In support of this hypothesis, we previously described, in another α-syn rat model of PD, that the main alterations occurred in the DAergic and glutamatergic terminals of the dorsolateral striatum and that these preceded SN dopamine neuron degeneration [71]. Alternatively, primarily degenerating DA neurons would cause their nerve endings to decrease their ability to transport GDNF to SN, thus worsening the neuropathology of the disease.

With both hypotheses being open and needing further investigation, our work has highlighted the importance of GDNF and contributes to the evidence of possible therapeutic use of this trophic factor in PD. Considering the difficulties in the direct administration of trophic factors, [72], possible strategies for PD might include non-pharmacological therapies such as direct current stimulation [73], not only because of its effects on degradation of α-synuclein, but also because of its possible role in determining an increase in trophic factors [74].

## 4. Materials and Methods

### 4.1. Genetic Animal Model

All animal experiments, which satisfied ARRIVE guidelines, were performed in accordance with European Communities Council Directive (2010/63 EU) as adopted by the Santa Lucia Foundation Animal Care and Use and approved by the Italian Ministry of Health (Authorization No. 617-2019PR). Homozygous bacterial artificial chromosome (BAC) transgenic rats (Sprague-Dawley background) overexpressing the full-length human *Snca* locus under the control of the endogenous human regulatory elements (*Snca^+/+^*) and wild type (WT) littermates of 5 and 12 months of age were used.

### 4.2. Histological and Immunohistochemical Studies

Tissue processing

The study groups included: 5-month-old wild type rats (WT), 5-month-old *Snca^+/+^* rats (*Snca^+/+^*) and 12-month-old WT and *Snca^+/+^* rats. Seven rats per group were used, and they were handled under the same conditions by one investigator at the same day and time. All rats were transcardially perfused under deep anesthesia with saline solution containing 0.01 mL heparin, followed by 4% paraformaldehyde (PFA) in phosphate buffer (PB; 0.1 M, pH 7.4). Brains were removed and post-fixed in 4% paraformaldehyde at 4 °C overnight and subsequently transferred into 30% sucrose solution at 4 °C until sinking. Sectioning was performed on a sliding cryostat (Leica, Wetzlar, Germany) at a thickness of 40 μm. Blinded observers collected primary data.

Histological and immunohistochemical studies

For the immunohistochemical studies, primary omission controls, normal mouse, and rabbit serum controls and preimmune serum controls were used to confirm the specificity of our immunohistochemical labeling.

Striatal neuronal cell count. Single-label immunofluorescence was performed using an antibody against Calbindin D-28K (marker of the Medium Spiny Neurons) (CALB, Immunological Sciences, Rome, Italy) to evaluate the number of surviving projection neurons in the striatum. Cell counts were carried out in each of 24 1.0-mm-square confocal microscope fields, rostrocaudally spaced on both hemispheres of 7 mice from each group. For cell counts, we used the Java image processing and analysis program Image J, developed by Wayne Rasband, available in the public domain (http://imagej.nih.gov/ij/docs/index.html (accessed on 1 January 2020). We used a manual approach, measuring the number of objects by means of the point selection tool.

Striatal interneuron subtype characterization

Immunohistological staining for striatal interneurons markers was performed. Brain sections were incubated with goat anti-choline acetyl transferase (ChAT; Novus Biological, Bio-Techne, Milan, Italy) and mouse anti-parvalbumin (PARV, Chemicon International, Inc., Temecula, CA, USA). All primary antibodies were used at a dilution of 1:200, in 0.1 M PB containing 0.3% Triton X-100 for 72 h at 4 °C. Sections were rinsed three times for 5 min at room temperature and subsequently incubated with secondary antibodies Alexa Fluor 488 and 555 for 2 h at room temperature at a dilution of 1:200 in a 0.1 M PB solution containing 0.3% Triton X-100. Confocal laser scanner microscopy (Zeiss, Oberkochen, Germany LSM800) was used to acquire images. Immunofluorescence analyses were performed using the Java image processing and plugin analysis program included in Fiji ImageJ.

Immunohistochemistry for GDNF

Peroxidase-antiperoxidase diaminobenzidine tetrahydrochloride single-label immunohistochemistry for GDNF was performed to identification and quantification of neurons involved in its production in the striatum. Serial sections were incubated with rabbit anti-GDNF (Novus Biological, Bio-techne, Milan, Italy) at a dilution of 1:100 in 0.1 M PB solution containing 0.3% Triton X-100 for 72 h at 4 °C. Subsequently, sections were incubated with rabbit peroxidase–antiperoxidase complex at a dilution of 1:100 in 0.1 M PB solution with 0.3% Triton X-100 at room temperature for 1 h. After peroxidase–antiperoxidase incubation, sections were incubated in Tris-Hcl buffer containing 10 mg diaminobenzidine tetrahydrochloride for 2 min, adding 15 µL of 3% hydrogen peroxidase. The peroxidase–antiperoxidase diaminobenzidine tetrahydrochloride-labeled sections were then washed in distilled water, placed in 0.1 M PB, mounted on gelatin-coated slides, dried, dehydrated and coverslipped. Collected images of GDNF positive neurons were obtained by Olympus U-RFLT 200 software (Olympus, Tokyo, Japan).

Analysis of colocalization of GDNF in striatal projection neurons, interneurons, and dopaminergic neurons

Double-label immunofluorescence was performed to evaluate the colocalization of GDNF in Calbindin-labeled striatal neurons, ChAT+, PV+ interneurons and TH+ dopaminergic neurons. Antigen retrieval was performed in Citrate Buffer (pH 6–7) for 20 min at 80 °C. Subsequently, sections were incubated with a cocktail of anti-GDNF, ChAT, PARV and TH (monoclonal mouse antibody, Millipore, Italy) antibodies for 72 h at +4 °C. A streptavidin–biotin amplification method for GDNF immunofluorescence staining was used. A confocal laser scanning microscope was used to acquire all the images. Three separate fields (dorsolateral, central, and medial each 1 mm in diameter) in each of three rostrocaudally spaced sections of six mice per group were examined. GDNF, Calbindin, ChAT, PV and TH immunofluorescence intensity was measured and quantified by using the Java image processing and analysis program available in, Fiji ImageJ. All confocal z-stacks images were acquired under no saturation conditions, with a ×63 objective raising a ×1 zoom with an XY resolution of 1024 × 1024 and 0.60 μm step-size along the vertical z-axis. The same set configuration was performed for all samples. Colocalization of GDNF in striatal projection neurons, interneurons, and dopaminergic neurons was calculated using the JACoP tool measures colocalization by performing a correlation coefficient-based analysis of the pixel intensity of different color channels. JACoP allows the use and comparison of some common intensity coefficient-based methods, including Pearson’s coefficient, Manders’ coefficient, Costes’s approach, and object-based analysis. The mean Mander’s coefficients were obtained by performing Costes Autothresholds on the samples’ regions of interest (ROIs) and running the analysis on 100 randomized images, and varied from 0 to 1, corresponding to non-overlapping images and 100% or quite colocalization between the two images, respectively.

### 4.3. Statistical Analysis

All the collected images were quantified by using the Java image processing and analysis program Fiji ImageJ. Cells of interest were selected using the freehand tool. From the Analyze menu, set measurements Mean “Gray Value”, “Area” and “Min and Max Gray Value” were selected. The region characterized by absence of fluorescence was considered in the background and it was subtracted. Finally, the mean values with SEM were obtained for all measures. ANOVA analysis available in the software GraphPad Prism version 9.0 was performed. p values < 0.05 were considered statistically significant.

## Figures and Tables

**Figure 1 ijms-23-10126-f001:**
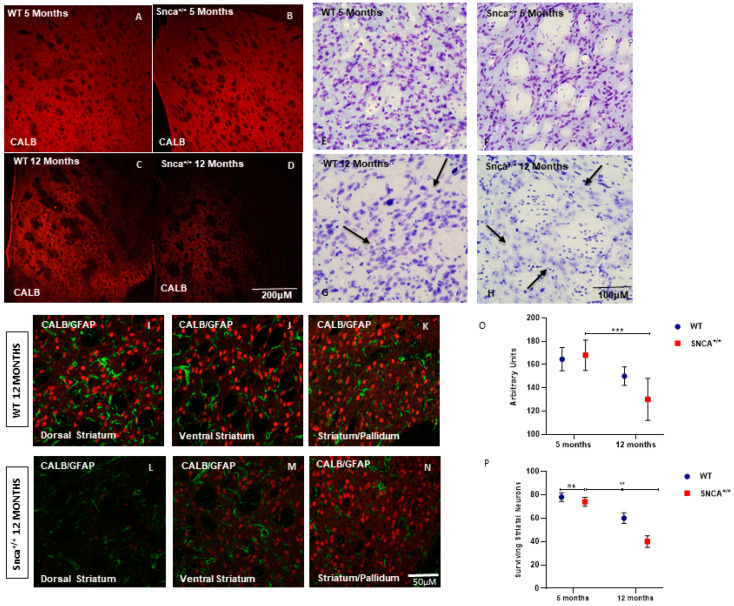
Striatum pathology. Confocal laser scanning microscopy images of single-label immunofluorescence for the marker of the medium spiny neurons, Calbindin D-28K (red), in the striatum of 5-month-old WT (**A**), 5-month-old *Snca^+/+^* (**B**), 12-month-old WT (**C**), and 12-month-old *Snca^+/+^* (**D**). Striatum Nissl staining. Representative images of coronal striatal slices (**E**–**H**) showing the dramatic reduction in surviving neurons in 12-month-old *Snca^+/+^* (*arrows*) compared to WT littermate (*arrows*). WT 12 months: (**I**) dorsal striatum, (**J**) ventral striatum, (**K**) striatum/pallidum; *Snca^+/+^* 12 months: (**L**) dorsal striatum, (**M**) ventral striatum, (**N**) striatum/pallidum; (**O**,**P**) Quantification of the number of striatal neurons labeled with CALB. A two-way ANOVA indicated a significant effect of genotype [F(1.24) = 7.86; ** *p* = 0.0098]; time [F(1.24) = 36.9 *** *p* < 0.001] and genotype X time interaction (F(1.24) = 3.49; *p* = 0.05). “ns” means “not significant”.

**Figure 2 ijms-23-10126-f002:**
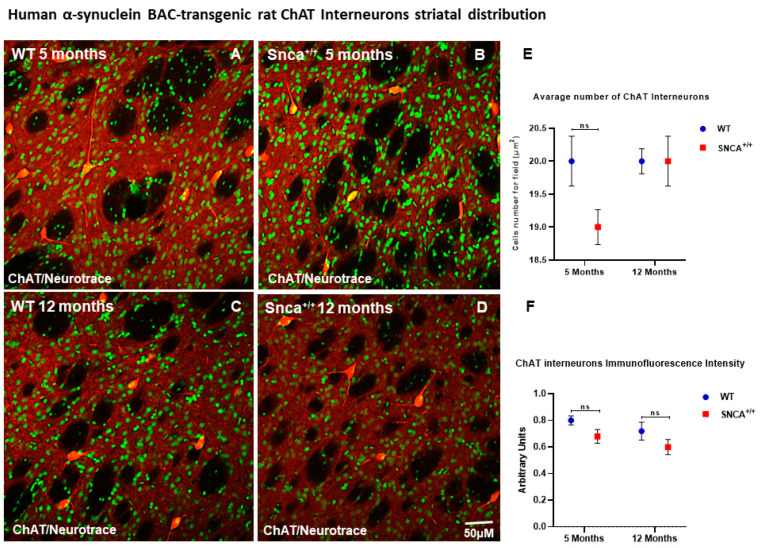
ChAT Interneuron Distribution. Confocal microscopy-acquired images of single-label immunofluorescence for ChAT and counterstained with fluorescent Nissl staining. ChAT is shown in red fluorescence, Neurotrace is labeled in green. (**A**–**D**) Images show the immunoreaction intensity of ChAT in each experimental group. (**E**,**F**) Two-way ANOVA analysis performed on data obtained from 5- and 12-month-old WT and *Snca^+/+^* showed no statistically significant (ns) change in ChAT-positive neurons and intensity.

**Figure 3 ijms-23-10126-f003:**
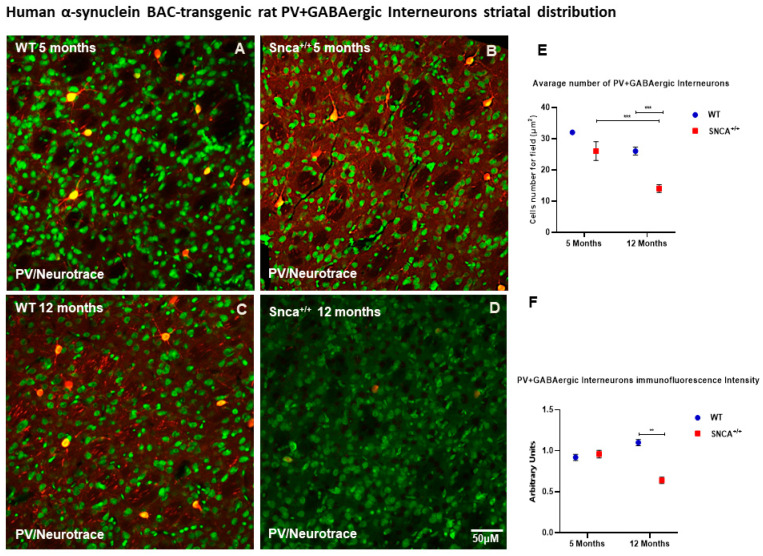
PV+ GABAergic distribution. Representative confocal microscopy-acquired images of single-label immunofluorescence for PV counterstained with Neurotrace. (**A**–**D**) Images show the shape and immunoreaction intensity of Pv+ GABAergic interneurons in each experimental group. (**E**,**F**) Two-way ANOVA analysis performed on data obtained from 5- and 12-month-old WT and *Snca^+/+^* showed a statistically significant reduction in number [genotype and time effect F(1.25) = 23.40 *** *p* < 0.001] and intensity [genotype effect F(1.207) = 10.80 ** *p* < 0.0012] of Parvalbumin-positive striatal neurons in 12-month-old *Snca^+/+^* rats.

**Figure 4 ijms-23-10126-f004:**
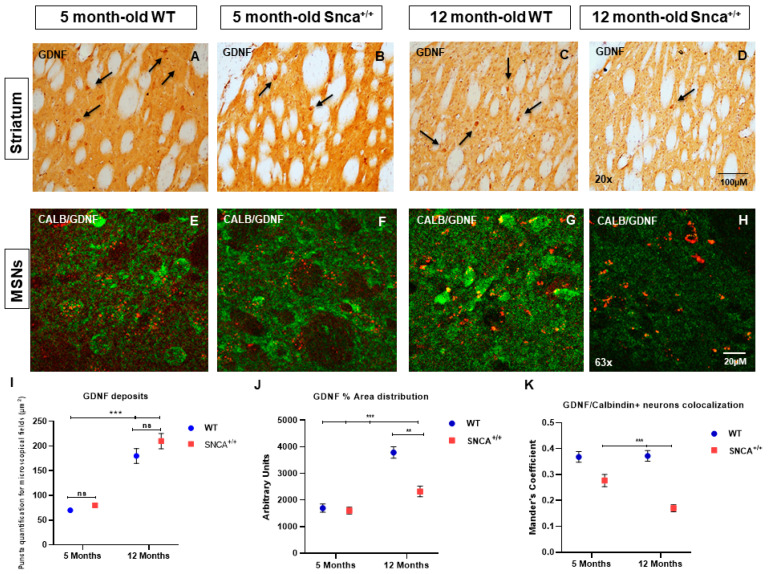
(**A**–**H**) GDNF in striatal projection neurons of WT and *Snca^+/+^* rats. Representative transmitted light microscope images showing Dab staining for GDNF counterstained with Hematoxylin in the striatum of 5- and 12-month-old Wt and *Snca^+/+^* rats. (*Arrows show the distribution of GDNF in the interneurons*). Two-way ANOVA revealed a statistically significant effect of Time (F1.157 = 115.1; *** *p* < 0.001) in GDNF deposit formation in the experimental groups of 5- and 12-month-old WT and *Snca^+/+^* rats, and a statistically significant genotype and time effect in GDNF area distribution [genotype F(1.157) = 17.84 ** *p* < 0.01; time F(1.157) = 57.99; *** *p* < 0.001]. “ns” means “not significant”. (**I**) GDNF deposits; (**J**) GDNF% area distribution; (**K**) GDNF/Calbindin + neurons colocalization.

**Figure 5 ijms-23-10126-f005:**
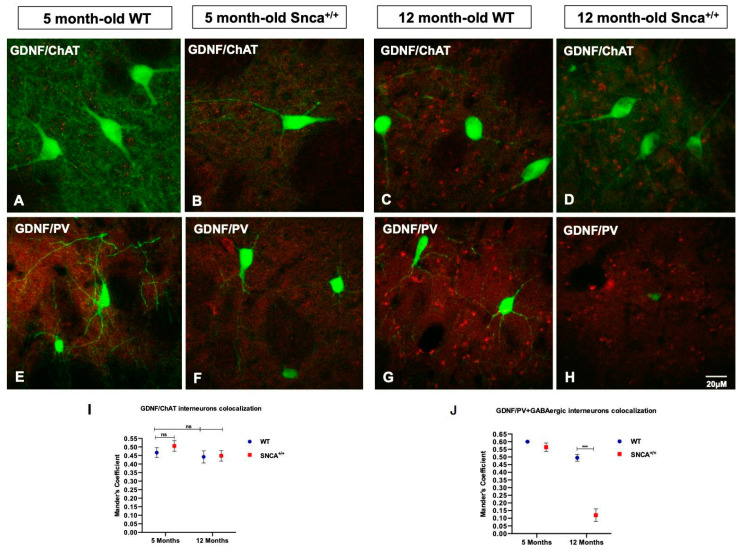
GDNF in striatal interneurons. (**A**–**D**) Representative confocal images show the immunoreaction intensity of ChAT and GDNF in each experimental group. (**E**–**H**) Images show the immunoreaction intensity of PV+ GABAergic neurons and GDNF in each experimental group. (**I**,**J**) Quantitative analysis of coexpression levels by Mander’s coefficient in all experimental groups. Histogram represents the mean of ROIs calculated on the basis of Mander’s coefficients, revealing the statistically significant colocalization of GDNF in ChAT- and PV-positive neurons, at *** *p* < 0.001, only in the 5-month-old WT and *Snca^+/+^* rats. “ns” means “not significant”.

**Figure 6 ijms-23-10126-f006:**
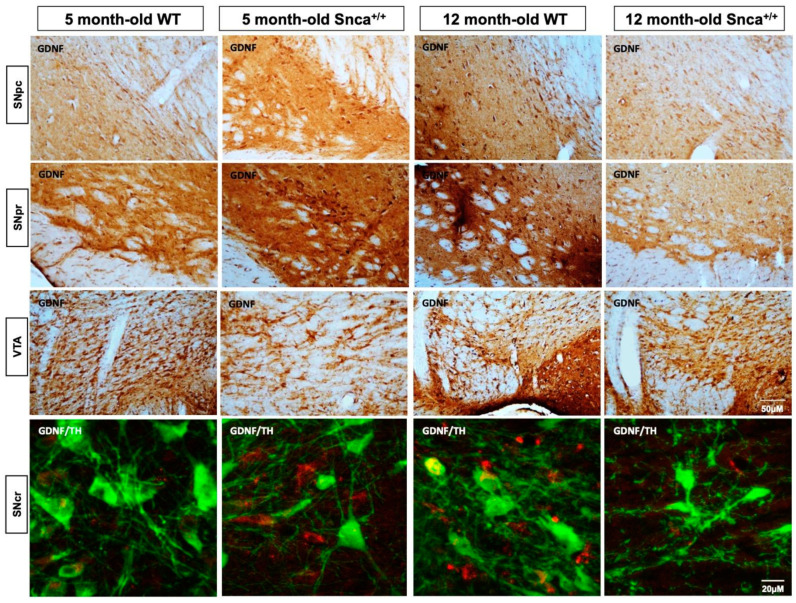
GDNF in the *Snca^+/+^* Substantia Nigra. Representative photomicrograph of DAB immunohistochemistry for the marker GDNF in rats Substantia Nigra. A qualitative intensity of colorimetric reaction is observed in the SNpr of the 12-month-old WT rats. These data are confirmed by confocal z-stack images of a double immunofluorescence for GDNF and TH in which colocalization analysis show a Mander’s coefficient greater than 0.7 in 5 and 12-month-old Wt that is significantly reduced in 12-month-old *Snca^+/+^*, in which it is less than 0.2.

## Data Availability

Not applicable.

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
