# Peer review of "Neuropathology of the Basal Ganglia in SNCA Transgenic Rat Model of Parkinson’s Disease: Involvement of Parvalbuminergic Interneurons and Glial-Derived Neurotropic Factor"

_ijms, 2022, doi:10.3390/ijms231710126_

Round 1
Reviewer 1 Report
This is a very well written paper, describing interesting and novel findings about the pathophysiology of Parkinson's Disease. I have only few minor concerns:
- my major concern is about the lack of a paragraph discussing both alternative explanations and pitfalls of this research. In this section, I think, two main points should be discussed in detail:
1) the significance of the reported findings considering other structures and nuclei critically involved in the pathophysiology of PD (i.e. the cerebellum; it is well known that the cerebellum itself plays a key role both in tremor onset and cognitive disturbances);
2) during disease progression, a critical role for the emergence of motor and non-motor fluctuations is commonly attributed to serotoninergic and noradrenergic axons, coming from locus coeruleus and raphe dorsalis nucleus respectively. Both these terminals do not express the pre-synaptic receptor DAT and can not storage dopamine; overall, the all-or-none release of dopamine from these axons mostly explain the development of motor and non-motor fluctuations. The Authors should explain whether and how their findings can match with these well-recognized mechanisms.
- in the Introduction section, the Authors did not consider another PD phenotype, clinically characterized by postural instability and gait disturbances at onset, now included in the classification of PD.
- the Authors should briefly explain how their findings could drive some recent and promising non-pharmacological approaches for the treatment of PD and other neurodegenerative disorders, characterized by the accumulation of pathological proteins (see Sala et al., Sci Rep 2021).
Author Response
Dear Reviewer, we have carefully read your criticisms and found it very useful to improve the manuscript.
1) the significance of the reported findings considering other structures and nuclei critically involved in the pathophysiology of PD (i.e. the cerebellum; it is well known that the cerebellum itself plays a key role both in tremor onset and cognitive disturbances);
We incorporated a paragraph in the discussion that involves the different circuits and neuroanatomical areas involved in PD, with appropriate references.
2) during disease progression, a critical role for the emergence of motor and non-motor fluctuations is commonly attributed to serotoninergic and noradrenergic axons, coming from locus coeruleus and raphe dorsalis nucleus respectively. Both these terminals do not express the pre-synaptic receptor DAT and can not storage dopamine; overall, the all-or-none release of dopamine from these axons mostly explain the development of motor and non-motor fluctuations. The Authors should explain whether and how their findings can match with these well-recognized mechanisms.
We incorporated a paragraph about this and again we thank you for this stimulating discussion
- in the Introduction section, the Authors did not consider another PD phenotype, clinically characterized by postural instability and gait disturbances at onset, now included in the classification of PD.
In the revised manuscript we now mention other models of PD including the aforementioned one.
- the Authors should briefly explain how their findings could drive some recent and promising non-pharmacological approaches for the treatment of PD and other neurodegenerative disorders, characterized by the accumulation of pathological proteins (see Sala et al., Sci Rep 2021).
In the last part of the discussion we have added a mention to novel therapeutic strategies to fight PD degeneration, including the reference Sala et al., Sci Rep 2021).
Reviewer 2 Report
Review of a manuscript: ”Neuropathology of the Basal Ganglia in SNCA transgenic rat model of Parkinson’s disease: involvement of parvalbuminergic interneurons and Glial-derived neurotropic factor” by Paldino and coauthors.
Parkinson’s disease is a severe neurodegenerative disease for which there is no treatment affecting the course of the disorder and no reliable biomarker for earlier identification. The authors studied neuropathological alterations in the striatum of a transgenic rat model of Parkinson’s disease. The other important direction of their research was evaluation of glial-derived neurotropic factor distribution in the striatum, and examination of striatum morphology. These are important directions of biomedical research and the results will be interesting for the readership of the journal.
The following corrections and additions should be done.
Abstract
Line 20. “A marked decrease in Glial derived neurotrophic factor,” The abbreviation of GDNF is already given on line 15 of Abstract, so it should be used here instead of the full name.
Introduction
Lines 28-29: ”The well-known PD triad of symptoms is characterized by tremor, rigidity, and bradykinesia and is linked to neuronal loss in the substantia nigra pars compacta.”The authors should add here a citation on a recent review of Parkinson’s disease [”Biomarkers in Parkinson’s Disease”. Chapter in a book Peplow P.V., Martinez B., Gennarelli T.A. (eds) Neurodegenerative Diseases Biomarkers. 2022. Neuromethods, vol 173. pp 155-180. Humana, New York, NY. https://link.springer.com/protocol/10.1007/978-1-0716-1712-0_7]
Results
1 Line 84: “Nissl staining performed on serial sections of rats brain tissue showed a significant neuronal depletion in the 12 months old Snca+/+ rats (Fig.1D).” The reference to figures in the text begins with Fig. 1D, but should with 1A. The authors should begin by description of Fig. 1A in the text or change the location of fragments on Figure 1 so the first reference should be on Figure 1A.
2 The authors also should explain more clearly what was a rational to use Snca+/+ rats in their experiments.
Discussion and elsewhere
Lines 229: ”PV neurons are evenly distributed throughout the Striatum”. Line 234 “In the striatum”.The authors should be consistent in writing striatum or Striatum.
Line 267: ”the importance of GDNF and contributes to the evidence of possible therapeutic use of this trophic factor in PD.” The authors should explain how they see a possible application of GDNF for the treatment of patients.
Supplementary Figure
The figures under the bar are too small, the fonts should be increased.
Overall: new interesting results, but some clarifications needed.
Author Response
Dear reviewer, thank you for your comments please find
Line 20. “A marked decrease in Glial derived neurotrophic factor,” The abbreviation of GDNF is already given on line 15 of Abstract, so it should be used here instead of the full name.
We have edited the text as suggested
Lines 28-29: ”The well-known PD triad of symptoms is characterized by tremor, rigidity, and bradykinesia and is linked to neuronal loss in the substantia nigra pars compacta.”The authors should add here a citation on a recent review of Parkinson’s disease [”Biomarkers in Parkinson’s Disease”. Chapter in a book Peplow P.V., Martinez B., Gennarelli T.A
We have identified a more appropriate part of the manuscript where we have added this interesting reference, which is in the discussion section
1 Line 84: “Nissl staining performed on serial sections of rats brain tissue showed a significant neuronal depletion in the 12 months old Snca+/+ rats (Fig.1D).” The reference to figures in the text begins with Fig. 1D, but should with 1A. The authors should begin by description of Fig. 1A in the text or change the location of fragments on Figure 1 so the first reference should be on Figure 1A.
We have edited the text as suggested
The authors also should explain more clearly what was a rational to use Snca+/+ rats in their experiments.
We have added a paragraph in the introduction following this suggestion
Lines 229: ”PV neurons are evenly distributed throughout the Striatum”. Line 234 “In the striatum”.The authors should be consistent in writing striatum or Striatum.
We corrected the issue
Line 267: ”the importance of GDNF and contributes to the evidence of possible therapeutic use of this trophic factor in PD.” The authors should explain how they see a possible application of GDNF for the treatment of patients.
The GDNF treatment is more extensively discussed in the revised version
The figures under the bar are too small, the fonts should be increased.
This has been corrected